# Stress Effects on Temperature-Dependent In-Plane Raman Modes of Supported Monolayer Graphene Induced by Thermal Annealing

**DOI:** 10.3390/nano11102751

**Published:** 2021-10-17

**Authors:** Yuehua Wei, Zhenhua Wei, Xiaoming Zheng, Jinxin Liu, Yangbo Chen, Yue Su, Wei Luo, Gang Peng, Han Huang, Weiwei Cai, Chuyun Deng, Xueao Zhang, Shiqiao Qin

**Affiliations:** 1College of Advanced Interdisciplinary Studies, National University of Defense Technology, Changsha 410073, China; 15116271019@163.com; 2College of Arts and Sciences, National University of Defense Technology, Changsha 410073, China; weizhsx@139.com (Z.W.); luoweihust@163.com (W.L.); 13507480737@126.com (G.P.); 3College of Physical Science and Technology, Xiamen University, Xiamen 361005, China; zhengxiaoming2020@xmu.edu.cn (X.Z.); jxliu@stu.xmu.edu.cn (J.L.); bobbychen@stu.xmu.edu.cn (Y.C.); suyuelnu@163.com (Y.S.); wwcai@xmu.edu.cn (W.C.); 4Hunan Key Laboratory of Super-Microstructure and Ultrafast Process, School of Physics and Electronics, Central South University, Changsha 410083, China; physhh@csu.edu.cn

**Keywords:** monolayer graphene, temperature-dependent in-plane Raman phonon modes, temperature coefficient, thermal annealing, compressive stress

## Abstract

The coupling strength between two-dimensional (2D) materials and substrate plays a vital role on thermal transport properties of 2D materials. Here we systematically investigate the influence of vacuum thermal annealing on the temperature-dependence of in-plane Raman phonon modes in monolayer graphene supported on silicon dioxide substrate via Raman spectroscopy. Intriguingly, raising the thermal annealing temperature can significantly enlarge the temperature coefficient of supported monolayer graphene. The derived temperature coefficient of G band remains mostly unchanged with thermal annealing temperature below 473 K, while it increases from −0.030 cm^−1^/K to −0.0602 cm^−1^/K with thermal annealing temperature ranging from 473 K to 773 K, suggesting the great impact of thermal annealing on thermal transport in supported monolayer graphene. Such an impact might reveal the vital role of coupling strength on phonon scattering and on the thermal transport property of supported monolayer graphene. To further interpret the thermal annealing mechanism, the compressive stress in supported monolayer graphene, which is closely related to coupling strength and is studied through the temperature-dependent Raman spectra. It is found that the variation tendency for compressive stress induced by thermal annealing is the same as that for temperature coefficient, implying the intense connection between compressive stress and thermal transport. Actually, 773 K thermal annealing can result in 2.02 GPa compressive stress on supported monolayer graphene due to the lattice mismatch of graphene and substrate. This study proposes thermal annealing as a feasible path to modulate the thermal transport in supported graphene and to design future graphene-based devices.

## 1. Introduction

As an important member in the family of two-dimensional (2D) atomic layer materials, graphene owns unique physical and chemical properties, which make it a promising candidate for various applications such as optoelectronic and high-frequency electronic devices [1,2,3,4,5,6]. However, the realization of these applications is hindered by several critical issues resulting from the fact that the thickness of graphene is much smaller than its average phonon mean free path (50~200 nm) [7,8,9,10]. Among those issues, power dissipation has become a constantly existing challenge on account of the improving packing density in integrated circuits nowadays, which puts a strict limitation on device performance and leads to data error or damages devices [7,10]. Practically speaking, graphene-based devices often operate under current saturation conditions and a huge amount of Joule heat is generated therein, resulting in a high local temperature and rigorous interaction between electron–phonon, phonon–phonon and the phonon interfaces [11]. As a matter of fact, the carrier mobility of graphene is largely restricted by the acoustic and optical phonon scattering [12]. Especially, the influence of phonon scattering on carrier mobility can be altered by local temperature, which largely depends on the heat dissipation efficiency of graphene devices. Thus, understanding the thermal transport characteristics of graphene is a necessary step in overcoming the power dissipation challenge in graphene devices.

Theoretical and experimental studies suggest that much effort has been put into power dissipation by designing a special pattern, introducing surface encapsulation and enhancing the coupling strength through current annealing [13,14]. Among them, thermal annealing has been used as an effective means to modulate the thermal transport characteristics [14,15]. Former literature reported that thermal annealing can bring strain and enhance the coupling strength between graphene and substrate [15], and the stress is closely related with coupling strength between graphene and substrate during the thermal annealing process. Besides, it is known that the sp^2^ bonds in graphitic carbon can endure extremely high mechanical strains and present interesting electromechanical properties [16]. Additionally, the remarkable strain effects on optical, electronic and thermal properties have been observed in honeycomb structure CNTs and graphene [16,17]. In general, the thermal transport including out-of-plane and in-plane properties can both be regulated by altering the strain resulting from the variation of coupling strength between graphene and substrate in theory and in experiments, which is highly affected by the residual H_2_O molecules at the graphene–substrate interface. The phonon dispersion can be modified by altering the coupling strength between graphene and substrate, especially for flexural acoustic (ZA) phonons, which are the main carriers of interfacial thermal transport (out-of-plane) in supported graphene [18]. In addition, the impact of coupling strength on in-plane thermal transport of graphene supported on an SiO_2_/Si substrate with an ultra-thin sild oxide layer (8 nm and 10 nm) has been experimentally characterized via an optothermal Raman technique [15]. As the coupling strength becomes stronger, the in-plane thermal conductivity (κ) gets reduced, which is ascribed to the enhancement of interface–phonon scattering and the mismatch of thermal expansion strains between graphene and substrate. As is known to all, the thermal transport in graphene can be depicted through the relationship of Raman peak positions versus ambient temperatures, and laser- or electrical-heating Raman thermometry is based on the acquisition of a temperature coefficient between Raman peak shift and temperature variation. In other words, the Raman peak shift induced by ambient temperature change can serve as an effective probe to comprehend the thermal expansion of lattice constant and phonon–interface scattering [19,20,21]. Therefore, a thermal annealing-related temperature-dependent Raman spectrum is vital for understanding the thermal transport and power dissipation of graphene devices. Although the thermal transport properties of graphene have been studied via Raman spectroscopy [22,23,24], there is limited literature on thermal annealing and temperature-dependent Raman spectrum for graphene supported on thicker SiO_2_ substrate (300 nm) so far.

In this work, the temperature-dependent Raman phonon modes were investigated for monolayer graphene supported on Si/SiO_2_ substrate (300 nm) after different thermal annealing processes via Raman thermometry. Specifically, the Raman peak position of supported monolayer graphene with no thermal annealing is redshifted, as the ambient temperature ranges from 193 K to 303 K, deriving the corresponding temperature coefficient of −0.030 cm^−1^/K. This may be attributed to the phonon softening during the ambient temperature increase. However, a discrepancy of temperature coefficient can be observed among supported monolayer graphene samples after vacuum thermal annealing at various annealing temperatures. For the lower annealing temperatures (373 K, 473 K), the temperature coefficients of annealed sample are comparable to that of pristine monolayer graphene. When the annealing temperature increases from 473 K to 773 K, a large discrepancy takes place in the temperature coefficients ranging from −0.030 cm^−1^/K to −0.0602 cm^−1^/K. The discrepancy could be attributed to the enhancement of strain induced by coupling strength and phonon–interface scattering. Moreover, the compressive stress on supported monolayer graphene through various thermal annealing processes are characterized by Raman spectroscopy. The compressive stress induced by thermal annealing exhibits a similar tendency, with the temperature coefficient under these thermal annealing processes, indicating the evident regulation from thermal annealing on thermal transport of supported monolayer graphene. After annealing at 773 K, the compressive stress runs up to 2.02 GPa, implying an enormous lattice mismatch between graphene and substrate. Our finding provides an insight into the thermal annealing of supported graphene and paves a new way to solve the thermal challenge in graphene-based devices.

## 2. Materials and Methods

The monolayer graphene was firstly prepared on SiO_2_/Si wafer with p-type-doping by traditional micromechanical exfoliated methods from the bulk graphite (Shanghai Onway Technology Co., Ltd., Shanghai, China). Characterized by optical microscopy (LV100D system, Nikon, Tokyo, Japan), the monolayer graphene was collected due to the discrepancy in optical contrast for graphene with different layers. The thickness of graphene was ulteriorly identified by Raman spectra (WITEC 300R Raman spectrophotometer). The temperature-dependent Raman spectra were also conducted (Renishaw spectrometer, Wotton-under-Edge, UK) with laser power of 1 mW and 50× long working objective. In the measurement, 532 nm exciting laser was focused, with the radius of 1 μm. The sample including graphene and a substrate located at a special seal cavity with transparent glass on the top and the temperature can be altered via the heat accessory holder (Linkam, Epsom, UK) with temperature accuracy ± 0.1 K. The temperature variation in the sample ranges from 303 K to 213 K through injecting the liquid nitrogen continuously. The monolayer graphene sample was annealed at the temperature ranging from 373 K to 773 K for 2 h in vacuum atmosphere (1.0 × 10^−1^ Pa) by quartz tube furnace (GSL-1500X-50). The temperature ramp was 2 °C/min in the beginning and the sample naturally cooled down to the room temperature, which can minimize the damage.

## 3. Results and Discussion

Graphene owns a typical honeycomb lattice structure [25], as illustrated in Figure 1a. The monolayer graphene flake can be seen from the optical image in Figure 1b, ascribed to the poor optical absorption of this atomic level thin graphene flake on Si/SiO_2_ substrate. After approximate identification of flake thickness by optical microscopy, Atomic force microscopy (AFM)and Raman spectroscopy to the red marked region in Figure 1b were also performed to get an accurate layer number of the graphene flake. According to the AFM image and height profile illustrated in Figure 1c, the flake thickness is about 0.32 nm, indicating that the graphene flake is monolayered. In the meantime, two obvious feature peaks located at 1584 cm^−1^ and 2673 cm^−1^ can be observed in the Raman spectrum (Figure 1d), assigned to in-plane vibrational G band and two phonon 2D bands, respectively. Based on the magnified 2D band in the inset of Figure 1d, the sharp and symmetric shaped 2D band can be fitted to one single Lorentz peak. Besides, the full width at half maximum (FWHM) of the examined 2D band is about 30 cm^−1^, in agreement with monolayer graphene characteristics [26]. In contrast, we also present the 2D band spectra of bilayer and trilayer graphene flakes in Appendix A. The FWHM for bilayer graphene is about 54 cm^−1^ while it is 59 cm^−1^ for trilayer graphene. The FWHM deviations of 2D band between these multiple layered graphene flakes are attributed to the variation in their electronic band structures. For this reason, the 2D band can be used as a sensitive probe to identify the number of graphene layers [27]. Furthermore, the thickness of graphene flake could also be distinguished by the intensity ratio of the 2D band against the G band [28,29]. Herein, it is found that the intensity of 2D band is twofold than that of G band, which testifies that the graphene flake is monolayered again. In a word, the graphene flake supported on Si/SiO_2_ substrate is monolayered judged by its optical image, AFM height profile and Raman spectrum.

To comprehend thermal annealing and thermal transport of supported monolayer graphene, temperature-dependent Raman spectroscopy was conducted for supported monolayer graphene flakes with various thermal annealing treatments (with thermal annealing temperature 373 K, 473 K, 673 K, 773 K). The Raman spectra in Figure 2 shows that the G band is shifted to a lower frequency as the ambient temperature ranges from 193 K to 303 K during Raman measurement, and the red shift of G band is universal for these diversely annealed graphene flakes. The red shift can be ascribed to phonon-softening induced by the increase of ambient temperature [30]. Comparison among Figure 2a–e reveals the alteration in Raman mode of supported monolayer graphene, which is clearly made by different thermal annealing temperatures. Overall, an obvious discrepancy of redshift can be seen in G band for the graphene flakes, which is the same case with the 2D band in Appendix A. Unlike pristine monolayer graphene, a larger redshift of the G band can be observed in the annealed monolayer graphene, due to the variation of strain and coupling strength between monolayer graphene and the substrate induced by thermal annealing. To understand the contribution of thermal annealing on thermal transport property, the values of G band redshift were summarized as a function of thermal annealing temperature illustrated in Figure 2f. The values for thermal annealing temperatures at 373 K and 473 K are comparable to the pristine value. When the annealing temperature increases from 473 K to 773 K, a large discrepancy occurs in the G band redshift, which uncovers the unusual influence of thermal annealing on thermal transport in supported monolayer graphene.

The Raman peak shifts are closely related to thermal transport in graphene. Former experimental studies show that the measurement of thermal transport, including in-plane thermal conductivity and interface thermal conductivity, is based on either 2D peak shift or G peak shift via Raman thermometry [2,9,23,24]. Thus, we investigate the Raman peak shifts of G band in detail to understand the thermal transport of supported monolayer graphene. Figure 3 shows the extracted G band positions versus Raman ambient temperatures for a series of thermal annealing graphene flakes. It can be observed that the G peak positions of all the supported monolayer graphene flakes are linear against Raman ambient temperatures. The relation of G band peak position versus ambient temperature may be linearly fitted as [7,31]
(1)$$\omega \left(T\right)=\omega_{0}+\chi T$$
where $\omega_{0}$ is the G peak position at 0 K and χ is the first-order temperature coefficient of the G band. The χ of G band is equal to the slope in fitting peak position and ambient temperature. It is noticed that some data points are different to the fitting curves in Figure 3a–e, which is possibly ascribed to the local temperature variation in the Raman cooling stage. The Raman temperature coefficient χ varies with thermal annealing temperature in Figure 3f. χ for 373 K and 473 K annealing processes are comparable to the pristine value. Nevertheless, a sharp rise in χ (−0.0602 cm^−1^/K) can be observed for 773 K thermal annealing temperature, which is twice than that of the unannealed pristine monolayer graphene (−0.03 cm^−1^/K). The reason for this tendency will be discussed in the following sections. The observed enhancement in Raman temperature coefficients may be contributed to by the anharmonicity scattering effect of phonon and the thermal expansion during ambient temperature increase [30]. The similar phenomenon has also been observed in χ of 2D band (Appendix A).

To pinpoint the root reason for the observed enhancement in temperature coefficient and to interpret the discrepancy induced by thermal annealing, we then attempt to investigate the thermal annealing mechanism from coupling strength of supported monolayer graphene and substrate. It is known that the thermal annealing can significantly enhance the degree of coupling strength for monolayer graphene supported on Si/SiO_2_ substrate due to the removal of H_2_O molecules. According to a previous report, the coupling strength is associated with strain [15]. Little strain results from the weak coupling strength during the thermal annealing process. As studied in previous reports [32,33], the water molecules from moisture in the atmosphere is easily trapped inside graphene and the substrate during mechanical exfoliation, resulting in the existence of water molecules at the interface between graphene and substrate. Generally, the interlayer water molecules between graphene and substrate exist as long as the graphene is mechanically exfoliated. Thus, there exist a few H_2_O molecules between graphene and substrate for the initially mechanical exfoliated graphene flake, as depicted in the schematic image (Figure 4a). Considering the lower adhesion energy of graphene/water/SiO_2_ compared to graphene/SiO_2_, these water molecules can be easily removed via thermal annealing. Thereby, a notable increase in coupling strength and strain would occur in the contact area between monolayer graphene and substrate, accompanied by the removal of water molecules and the enhancement of interfacial-phonon scattering. Except for strain, the ability of p-type charge carrier transfer and substrate doping also decreases during the thermal annealing process, as ascribed to the reduced reactivity of graphene. Based on the discussion mentioned above, strain, p-type charge carrier density or substrate doping may be the main reason for the enhancement of temperature coefficient. To completely exclude the influence of the substrate doping effect, we characterized the temperature-dependent Raman spectroscopy to monolayer graphene supported on a BN substrate, as presented in Appendix A. The temperature coefficient of monolayer graphene on BN substrate (0.031 cm^−1^/K) is comparable to that on Si/SiO_2_ substrate (0.030 cm^−1^/K), indicating that the temperature coefficient shows no obvious reliance on the type of substrate. Additionally, previous literature shows that the ratio of Raman 2D and G peak shift induced by thermal annealing can serve as an effective probe to reveal the roles of p-type charge carrier density and strain [15]. Thus, we also plotted the Raman peak positions of G and 2D band as a function of ambient temperature in Figure 4b and the two distinctive slopes are consistent with previous reports [34,35]. Appendix A summarizes the extracted temperature coefficients of Raman G and 2D band for supported monolayer graphene flakes upon different thermal annealing processes, respectively. The ratio of temperature coefficient for G and 2D band ($\frac{\Delta \omega_{2D}}{\Delta \omega_{G}}$) is approximately 2, which does not change with various thermal annealing processes (annealing at 373 K, 473 K, 673 K, 773 K), indicating that strain is the dominant role here, instead of p-type charge carrier density [15].

Additionally, the rising of temperature coefficient is believed to be ascribed to strain instead defects upon 673 K and 773 K annealing, because there exists little or no D peak in the Raman spectrum (Appendix A). As reported in a previous study, the intrinsic anharmonicity and thermal expansion strain make joint efforts for the Raman peak shift induced by thermal annealing [36]. Thus, we have $\Delta \omega^{\mathrm{total}}=\Delta \omega^{\mathrm{instrinsic}}+\Delta \omega^{\mathrm{strain}}$, as illustrated in Figure 4c. For monolayer graphene flakes annealed at 373 K and 473 K, the contribution of strain on Raman peak shifts is small, indicating weak coupling strength between monolayer graphene and substrate. For those, the Raman peak shifts are mainly ascribed to intrinsic anharmonicity. In other words, the relatively lower thermal annealing temperature plays a negligible role on the coupling strength and generates little or no strain. In contrast, the higher thermal annealing temperatures (673 K and 773 K) bring great enhancement in temperature dependency of Raman peak shift, which should be ascribed to the additional impact from $\Delta \omega^{\mathrm{strain}}$ besides $\Delta \omega^{\mathrm{instrinsic}}$. This tendency is the same as that in Figure 2f and Figure 3f. Theoretical research testifies that the enlarged temperature coefficient arises from the opposite thermal expansions of graphene (negative) and the substrate (positive) [15].

In order to identify the strain upon different thermal annealing processes, we have systematically analyzed the G band of supported monolayer graphene flakes after thermal annealing. The evolution of G band in different thermal annealing processes is shown in Figure 5a–d. An obvious finding is that the G bands of all the supported monolayer graphene flakes are shifted to higher frequencies after annealed in a vacuum. Furthermore, a larger discrepancy of blue shift can be observed for supported monolayer graphene as the thermal annealing temperature increases. The G band blue shift keeps almost unchanged upon lower thermal annealing temperatures (373 K, 473 K), while it rises sharply as the thermal annealing temperature rises from 473 K to 773 K. According to the literature and discussion above, the blue shift of the G band can be attributed to compressive stress or p-type charge doping upon the thermal annealing process [16]. To tell the true mechanism of thermal annealing, we also analyze the bandwidth of G band. As shown in Figure 5a–d, the bandwidth also fluctuates around 3 cm^−1^, which is smaller than the bandwidth of 10 cm^−1^ induced by charge doping [37,38]. Thus, the charging doping should be excluded in our experiment, which is the same as was discussed earlier in Figure 4. On the other hand, the 2D band blueshift is further evidence to identify the presence of p-type charge doping or substrate doping, as reported by Yan et al. [38] and Pisana et al. [37] It was found that there was a very small influence in p-type charge doping on 2D band blueshift. However, the 2D band is blueshifted by ~15 cm^−1^ in Appendix A in our experiment. Consequently, the large blueshift induced by thermal annealing cannot be attributed to p-type charge doping or substrate doping. Thus, the compressive stress should be the main reason for the blueshift of G and 2D band in supported monolayer graphene upon different thermal annealing temperatures.

The compressive stress is originated from the lattice mismatch at graphene and substrate interface induced by thermal annealing. Therefore, the generated stress should be biaxial. By analyzing the blue shift of Raman G band, the value of biaxial compressive stress can be estimated. In hexagonal system, the strain ϵ induced by biaxial stress σ can be described with the following equation [16,39,40],
(2)$$\left[\begin{array}{c} \varepsilon_{xx}\\ \varepsilon_{yy}\\ \varepsilon_{zz}\\ \varepsilon_{yz}\\ \varepsilon_{zx}\\ \varepsilon_{xy} \end{array}\right]=\left[\begin{array}{cccccc} S_{11} & S_{12} & S_{13} & 0 & 0 & 0\\ S_{12} & S_{11} & S_{13} & 0 & 0 & 0\\ S_{13} & S_{13} & S_{33} & 0 & 0 & 0\\ 0 & 0 & 0 & S_{44} & 0 & 0\\ 0 & 0 & 0 & 0 & S_{44} & 0\\ 0 & 0 & 0 & 0 & 0 & 2S_{11}-S_{12} \end{array}\right]\left[\begin{array}{c} \sigma \\ \sigma \\ 0\\ 0\\ 0\\ 0 \end{array}\right]$$

Considering the *x* and *y* axes in graphene plane and the *z* axis perpendicular to the plane, we get $\varepsilon_{xx}=\varepsilon_{yy}=\left(S_{11}+S_{12}\right)\sigma$, $\varepsilon_{zz}$ = 2$S_{13}\;\sigma$, $\varepsilon_{yz}=\varepsilon_{zx}=\varepsilon_{xy}=0$. As the shear strain equals to zero, the secular equation can be simplified as
(3)$$\left|\begin{array}{cc} A\left(\varepsilon_{xx}+\varepsilon_{yy}\right)-\lambda & B\left(\varepsilon_{xx}-\varepsilon_{yy}\right)\\ B\left(\varepsilon_{xx}-\varepsilon_{yy}\right) & A\left(\varepsilon_{xx}+\varepsilon_{yy}\right)-\lambda \end{array}\right|=0$$
where $\lambda =\omega_{\sigma }^{2}-\omega_{0}^{2}$, $\omega_{\sigma }$ and $\omega_{0}$ is the frequencies of Raman $E_{2g}$ phonon (G mode) upon stressed and unstressed conditions. The solution for this equation can be written as
(4)$$\lambda =A\left(\varepsilon_{xx}+\varepsilon_{yy}\right)=2A\varepsilon_{xx}=2A\left(S_{11}+S_{12}\right)\sigma$$

Thus,
(5)$$\omega_{\sigma }-\omega_{0}=\frac{\lambda }{\omega_{0\tau }+\omega_{0}}\approx \frac{\lambda }{2\omega_{0}}=\frac{A\left(S_{11}+S_{12}\right)\sigma }{\omega_{0}}=\alpha \sigma$$
where α is the stress coefficient. Since $A=-1.44\times 10^{7}{\mathrm{cm}}^{-2}$ and graphite elastic constants $S_{11}=0.98\times 10^{-12}{\mathrm{Pa}}^{-1}$ and $S_{11}=0.16\times 10^{-12}{\mathrm{Pa}}^{-1}$, and $\omega_{0}=1580{\;\mathrm{cm}}^{-1}$, the estimated stress coefficient α is ~7.47 cm^−1^/GPa. Figure 5f shows the calculated compressive stress of supported monolayer graphene upon different thermal annealing process according to the estimated α. This result clearly indicates that raising annealing temperature can lead to the enhancement of compressive stress. As the monolayer graphene anneals at 773 K, the generated compressive stress is appropriately 2.02 GPa, which is much larger than that of pristine monolayer graphene. The calculated compressive stress is close to a previous report [16]. Such a phenomenon proves that the enhanced compressive stress has a great influence on the thermal transport characteristics of supported monolayer graphene, as discussed in a previous section. However, the compressive stress at lower thermal annealing temperature stages (373 K, 473 K) shows a slower variation, sharing the same tendency of temperature coefficient variation in our experiment.

The relationship between compressive stress and annealing temperature can be fitted as:(6)$$\sigma =-0.0155+\left(2.36\times 10^{-3}\right)T+\left(5.17\times 10^{-6}\right)T^{2}$$
where σ and T represent the compressive stress and temperature, respectively. This enhanced compressive stress upon thermal annealing mainly comes from the fact that monolayer graphene is easily compressed or expanded owing to the ultra-thin thickness of monolayer graphene (0.325 nm). It is worth noting that the compressive stress induced by thermal annealing can improve the thermal transport of monolayer graphene significantly. These results are similar to our earlier discussion on the variation of temperature coefficient, further demonstrating the important compressive stress effect on thermal transport of graphene. Therefore, stress should be the main reason affecting the thermal transport of monolayer graphene upon different annealing temperature.

## 4. Conclusions

In summary, we systematically investigated the thermal annealing temperature-dependent phonon modes of monolayer graphene supported on substrate by Raman spectroscopy with ambient temperature ranging from 193 K to 303 K. As the ambient temperature rises up, the Raman peak position of G mode is redshifted due to phonon softening. Moreover, such a redshift is observed commonly for monolayer graphene samples upon different thermal annealing treatment. Especially for 773 K annealing, the temperature coefficient suddenly surges to −0.0602 cm^−1^/K, which is twofold than that of pristine monolayer sample without any thermal annealing. The influence from thermal annealing on the temperature coefficient of supported monolayer graphene might be attributed to the increased coupling strength and enhanced compressive stress. As an evidence, the corresponding Raman analysis also shows that the compressive stress in annealed monolayer graphene can reach up to 2.02 GPa. Our finding proposes thermal annealing as an alternative route to regulate the thermal transport and heat dissipation of supported graphene and other 2D material devices.

## Figures and Tables

**Figure 1 nanomaterials-11-02751-f001:**
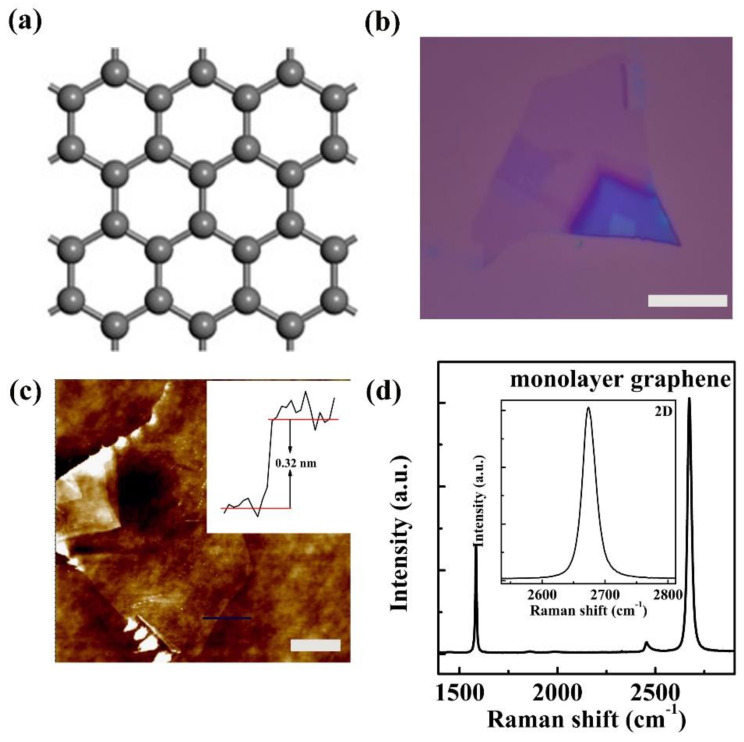
Basic characterizations of supported monolayer graphene. (**a**) Crystalline structure; (**b**) Optical image; the scale bar is about 10 μm. (**c**) AFM image, inset is its affiliated height profile and the scale bar is about 10 μm; (**d**) The Raman spectrum, inset is the magnified Raman 2D band.

**Figure 2 nanomaterials-11-02751-f002:**
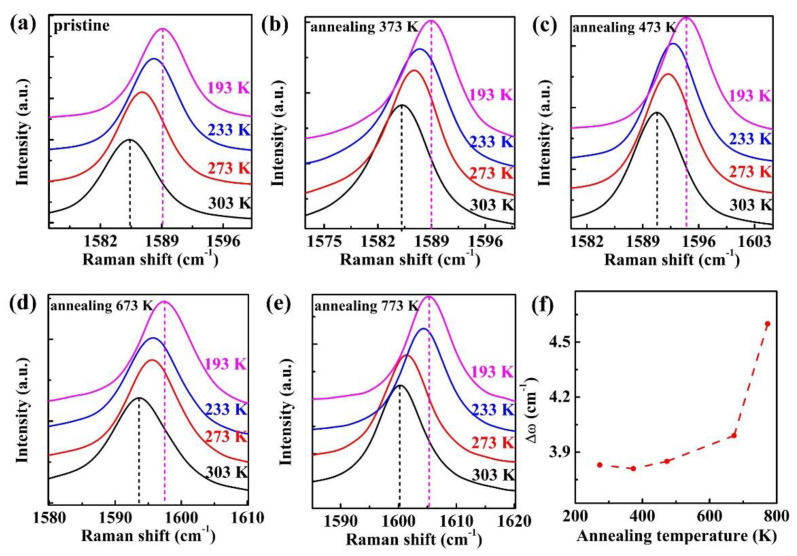
The temperature-dependent Raman spectra of G band for supported monolayer graphene flakes with various thermal annealing processes. (**a**) Pristine; (**b**) Annealing at 373 K; (**c**) Annealing at 473 K; (**d**) Annealing at 673 K; (**e**) Annealing at 773 K; (**f**) The evolution of G band redshift as a function of thermal annealing temperature.

**Figure 3 nanomaterials-11-02751-f003:**
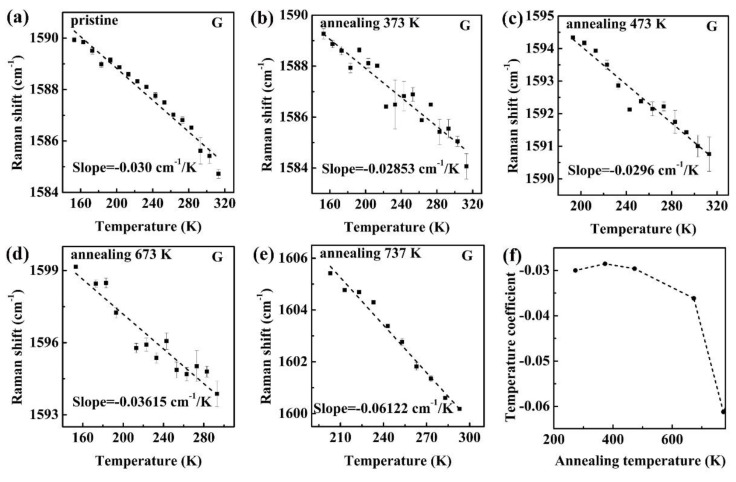
The temperature coefficients of supported monolayer graphene flakes with various thermal annealing processes. (**a**) Pristine; (**b**) Annealing at 373 K; (**c**) Annealing at 473 K; (**d**) Annealing at 673 K; (**e**) Annealing at 773 K; (**f**) The extracted temperature coefficients of monolayer graphene under different annealing temperatures.

**Figure 4 nanomaterials-11-02751-f004:**
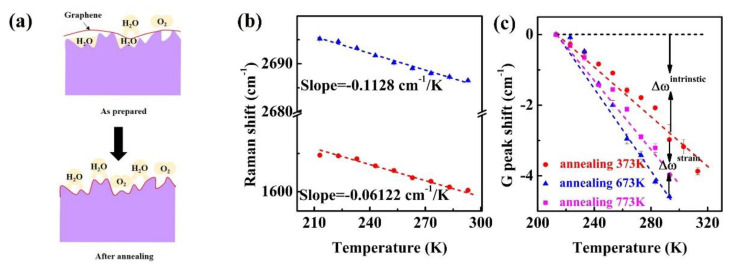
(**a**) The schematic image about the thermal annealing mechanism in monolayer graphene supported on Si/SiO_2_ substrate; (**b**) The temperature-dependence of Raman G and 2D peak position, blue data points belonging to 2D peak and red data points to G peak; (**c**) The temperature-dependent Raman peak shifts of G band under different thermal annealing processes.

**Figure 5 nanomaterials-11-02751-f005:**
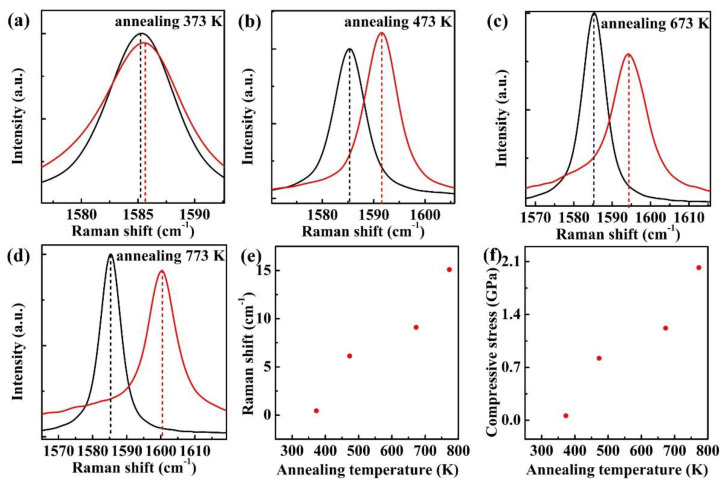
The G band Raman spectra measured at room temperature of supported monolayer graphene after thermal annealing at different temperatures. (**a**–**d**) The Raman spectra annealing at 373 K, 473 K, 673 K, 773 K; (**e**) The relationship between G band blue shift and thermal annealing temperature; (**f**) The calculated compressive stress on monolayer graphene after different thermal annealing processes.

## Data Availability

The data presented in this study are available on request from the corresponding author.

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
