# Peer review of "Stress Effects on Temperature-Dependent In-Plane Raman Modes of Supported Monolayer Graphene Induced by Thermal Annealing"

_nanomaterials, 2021, doi:10.3390/nano11102751_

Round 1
Reviewer 1 Report
The authors reported the stress effects on temperature-dependent in-plane Raman modes of monolayer graphenes supported onto SiO2/Si substrates. It is very afraid to mention that this work seems to be not quite new one providing little critical insight into the community since many studies on the temperature dependent Raman spectroscopy on a monolayer graphene addressing stress effects have been extensively reported so far [eg., https://doi.org/10.1016/j.carbon.2017.09.033, https://doi.org/10.1073/pnas.0811754106, https://doi.org/10.1007/s12274-008-8036-1, https://doi.org/10.1021/nl071033g, https://doi.org/10.1021/acs.jpclett.1c00947, etc.] Thus, it is not clear where the novelty of this work might be. Nevertheless, the result for high-temperature thermal treatment (up to 773 K in this work) on monolayer graphene would be an interesting point for consideration on possible publication if the authors would clearly address the critical issues raised below.
- Results on Fig. 2, 3, and 4 as well as their discussion appear to be quite similar to the Ref. 15 (https://doi.org/10.1016/j.carbon.2017.09.033) cited in this work, thus little information would be (especially for Fig. 4).
- The authors did not fully provide the specific information on their supporting substrates, SiO2/Si, e.g. doping type of Si wafer. They argued that vacuum-annealing treatment on graphene resulted in enhancement of the coupling strength between graphene and substrate because intercalated water molecules would be dehydrated as annealing temperature increased, which was previously already evidenced by in Ref. 15 shown above. But, this issue is not clearly resolved in this work as the authors did not fully exclude the substrate doping effect, which is a critical resource for Raman shift (e.g., https://pubs.acs.org/doi/abs/10.1021/jp8008404, https://doi.org/10.1038/nchem.1421, https://doi.org/10.1007/s12274-008-8036-1).
- In the context mentioned above, it is kindly recommended for the authors to carried out the same temperature-dependent Raman spectroscopy on either suspended graphene or graphene on different substrates to fully exclude substrate doping effect.
- In Ref. 15, the graphene used was wet-transferred onto the SiO2/Si substrate providing the chance of existence of the water molecules being remain between graphene and the underlying substrate. But, in this work, the graphene used was first mechanically exfoliated then dry-transferred onto the substrate with little chance of having water molecules compared to the one in Ref. 15. How would this issue be addressed?
Author Response
Response Letter
Journal: Nanomaterials
Manuscript ID: 1400119
Title: Stress effects on temperature-dependent in-plane Raman modes of supported monolayer graphene induced by thermal annealing
Dear editors and reviewers:
First of all, we would like to thank all the editors and reviewers for reviewing our manuscript and pointing out valuable comments, which are helpful for improving the manuscript and guiding further investigation. Based on the constructive comments and suggestions, we have revised the original manuscript. Below are the detailed replies, and the revisions are highlighted correspondingly in the modified manuscript.
Reply to comments of Reviewer 1:
Review 1
- Results on Fig. 2, 3, and 4 as well as their discussion appear to be quite similar to the Ref. 15 (https://doi.org/10.1016/j.carbon.2017.09.033) cited in this work, thus little information would be (especially for Fig. 4).
Reply: We thank the referee for the comment. The main topic in Ref.15 concentrates on the influence of different annealing process on thermal conductivity. Nevertheless, specific information about temperature-dependent in-plane Raman modes upon different annealing processes can hardly be obtained from Ref. 15. Additionally, the thickness of SiO2 layer is approximately 8 nm to 10 nm in Ref.15 while it is 300 nm in our work, which is more beneficial for electrostatic modulation in devices. The vital information in our context is the temperature-dependence of in-plane Raman phonon modes upon vacuum thermal annealing in quartz tube furnace. Previous literatures [Nano Letters 2017, 17 (6), 3429-3433; ACS Applied Materials & Interfaces 2018, 10 (29), 24892-24898; 2D Materials 2018, 5 (2), 025009; ACS Nano 2014, 8 (1), 986-993; Advanced Materials 31.7 (2019): 1804979] show that both the in-plane and interface thermal transport are based on the temperature coefficient calculated from the temperature-dependent Raman spectroscopy. Thereby, our manuscript is useful to subsequent study of the annealing effect on either in-plane or interface thermal transport.
- The authors did not fully provide the specific information on their supporting substrates, SiO2/Si, e.g. doping type of Si wafer. They argued that vacuum-annealing treatment on graphene resulted in enhancement of the coupling strength between graphene and substrate because intercalated water molecules would be dehydrated as annealing temperature increased, which was previously already evidenced by in Ref. 15 shown above. But, this issue is not clearly resolved in this work as the authors did not fully exclude the substrate doping effect, which is a critical resource for Raman shift (e.g., https://pubs.acs.org/doi/abs/10.1021/jp8008404, https://doi.org/10.1038/nchem.1421, https://doi.org/10.1007/s12274-008-8036-1).
Reply: Thank the referee very much for pointing out this issue. We are very sorry for not giving the specific information on SiO2/Si substrate. The doping type of Si wafer been used is p-type. The prior literatures [Nature chemistry, 2012, 4(9): 724-732; The Journal of Physical Chemistry C, 2008, 112(29): 10637-10640] report that the substrate doping effect mainly includes electron and hole doping. The electron doping has contribution to the redshift in graphene flakes while the hole doping gives rise to blueshift, indicating opposite displacement in Raman frequency. In our experiment, a diversity of blueshift in G and 2D band has been observed in graphene flakes upon different annealing processes. Because doping in 2D band is usually 10~30% weaker than in G band, the observed 2D band blueshift (~15 cm-1) is too large to be induced by hole doping. Thus, the substrate doping effect could be ruled out.
- In the context mentioned above, it is kindly recommended for the authors to carried out the same temperature-dependent Raman spectroscopy on either suspended graphene or graphene on different substrates to fully exclude substrate doping effect.
Reply: Thanks for the contributive advice. To completely exclude the influence of substrate doping effect, we characterized the temperature-dependent Raman spectroscopy to monolayer graphene supported on BN substrate, as presented in Fig. R1. The temperature coefficient of monolayer graphene on Si/SiO2 substrate (0.030 cm-1/K) is comparable to that on BN substrate (0.031 cm-1/K), indicating that the temperature coefficient shows no obvious reliance on the type of substrate. With these results, the influence of substrate doping on temperature-dependent Raman spectroscopy can be ruled out fully.
Fig. R1 The temperature-dependent Raman spectra of monolayer graphene on BN.
- In Ref. 15, the graphene used was wet-transferred onto the SiO2/Si substrate providing the chance of existence of the water molecules being remain between graphene and the underlying substrate. But, in this work, the graphene used was first mechanically exfoliated then dry-transferred onto the substrate with little chance of having water molecules compared to the one in Ref. 15. How would this issue be addressed?
Reply: Thank the referee very much for pointing out this issue. In our manuscript, the graphene with high quality is directly prepared on 300 nm SiO2/Si substrate via typical mechanical exfoliation method without dry-transfer process. As studied in previous reports [Nano Res. 2012, 5(10): 710–717; Nano Res. 2015, 8(9): 3020–3026], the water molecules from moisture in the atmosphere is easily trapped inside graphene and substrate during mechanical exfoliation, resulting the existence of water molecules at the interface between graphene and substrate. Generally, the interlayer water molecules between graphene and substrate exist as long as the graphene is mechanically exfoliated.
We hope that our response and revision have addressed reviewers’ comments and help to make our manuscript qualified for publication in Nanomaterials.
Sincerely yours,
Prof. Xueao Zhang (on behalf of all authors)
College of Physical Science and Technology, Xiamen University, Xiamen, 361005, China
Tel: 13875945032
E-Mail: xazhang@xmu.edu.cn

Reviewer 2 Report
The authors report a temperature dependent study of Raman emission from graphene
The results are interesting for researchers working in the field of graphene optoelectronics, but a few revisions must be addressed
- The author must include more details on the temperature dependent measurements. How were performed the measurements? Was the graphene or its substrate heated? Is the process reversible?
- The laser power (1 mW) can be huge (see A.C. Ferrari, Phys. Rev. Lett. 97 (2006) 1–4) for this type of measurements. Has the laser induced heating considered in the analysis and how?
- Since the annealing was performed at medium vacuum (10-1 Pa), it can be possible that some oxygen incorporation took place. Can the author exclude the formation of defects (like local oxidation) during the thermal annealing? Can they compare the PL emission before and after the annealing procedure?
Author Response
Response Letter
Journal: Nanomaterials
Manuscript ID: 1400119
Title: Stress effects on temperature-dependent in-plane Raman modes of supported monolayer graphene induced by thermal annealing
Dear editors and reviewers:
First of all, we would like to thank all the editors and reviewers for reviewing our manuscript and pointing out valuable comments, which are helpful for improving the manuscript and guiding further investigation. Based on the constructive comments and suggestions, we have revised the original manuscript. Below are the detailed replies, and the revisions are highlighted correspondingly in the modified manuscript.
Reply to comments of Reviewer 2:
Review 2
- The author must include more details on the temperature dependent measurements. How were performed the measurements? Was the graphene or its substrate heated? Is the process reversible?
Reply: We thank the referee for the comment. In our experiment, the temperature-dependent Raman spectroscopy was characterized via a confocal micro-Raman spectroscope (Renishaw inVia Qontor, UK). The sample including graphene and substrate located at special seal cavity with transparent glass on the top and the temperature can be altered via the heat accessory holder (Linkam, UK) with temperature accuracy ± 0.1K. The temperature variation in the sample ranges from 303 K to 213 K through injecting the liquid nitrogen continuously. Benefiting from the low temperature environment, no damage can be seen in graphene, leading to the reversible process in temperature-dependent Raman measurement. The corresponding details are added in the manuscript with red mark.
- The laser power (1 mW) can be huge (seeC. Ferrari, Phys. Rev. Lett. 97 (2006) 1–4) for this type of measurements. Has the laser induced heating considered in the analysis and how?
Reply: Thank the referee very much for pointing out this issue. The temperature-dependent Raman spectra were acquired using a Renishaw spectrometer with the wavelength of 532 nm. A 50x lens with 0.9 mm working distance can only be used for the temperature-induced Raman measurement. For better signal-to-noise ratio, at least 1 mW of laser power should be applied. We have conducted temperature-dependent Raman measurement on pristine monolayer graphene with laser power of 1 mW and 0.1mW to address the influence of laser power on Raman shift, as shown in Fig. R2. The results show little discrepancy in temperature coefficients for different laser power, indicating that laser induced heating from 1mW laser is negligible.
Fig. R2 The temperature-dependent Raman spectra of 1L (a) and 2L (b) graphene.
- Since the annealing was performed at medium vacuum (10-1 Pa), it can be possible that some oxygen incorporation took place. Can the author exclude the formation of defects (like local oxidation) during the thermal annealing? Can they compare the PL emission before and after the annealing procedure?
Reply: Thanks for the referee’s comment. Due to the gapless property of graphene, the PL emission is inexistent. However, the information of defect can be determined via the typical Raman spectroscopy characterization. Raman spectroscopy can provide useful information on G band, 2D band, as well as the defect band (D band). In contrast to TMDs, the defects of graphene can be identified by the D band in Raman spectrum. To explore the local oxidation, Raman mapping was conducted in thermal annealed graphene, as illustrated in Fig. R3 (b, c). Raman mapping enables a statistical analysis of homogeneity in monolayer graphene to determine whether thermal annealing plays a same role on the graphene flake. The uniform 2D band image(Fig. R3b) shows the spatial homogeneity of thermal annealing over the whole graphene flake. More importantly, there exists no D band signal at any position, and this excludes the oxygen incorporation during thermal annealing.
Fig. R3 The typical Raman characterization of monolayer graphene. (a) optical image; (b) Raman mapping of 2D mode; (c) Raman mapping of D mode; (d) Raman spectrum of pristine graphene; (e) Raman spectrum of graphene on 773 K annealing.
We hope that our response and revision have addressed reviewers’ comments and help to make our manuscript qualified for publication in Nanomaterials.
Sincerely yours,
Prof. Xueao Zhang (on behalf of all authors)
College of Physical Science and Technology, Xiamen University, Xiamen, 361005, China
Tel: 13875945032
E-Mail: xazhang@xmu.edu.cn

Round 2
Reviewer 1 Report
It seems that the authors were trying to do their best to answer all of my comments and questions. Now I agree with their claims on the originality of this manuscript, which are 'temperature dependence of in-plane Raman phonon modes upon vacuum thermal annealing'. Nevertheless, their claims on the originality as well as other critical answers and additional experiment on graphene on BN substrate have NOT been still incorporated into the revised manuscript at all. The authors are strongly recommended that all their comments and answers should be added in the 2nd revised manuscript as much as possible for the final acceptance of the manuscript.
Author Response
It seems that the authors were trying to do their best to answer all of my comments and questions. Now I agree with their claims on the originality of this manuscript, which are 'temperature dependence of in-plane Raman phonon modes upon vacuum thermal annealing. Nevertheless, their claims on the originality as well as other critical answers and additional experiment on graphene on BN substrate have NOT been still incorporated into the revised manuscript at all. The authors are strongly recommended that all their comments and answers should be added in the 2nd revised manuscript as much as possible for the final acceptance of the manuscript.
Thank the review very much. The claims and additional experiment of graphene on BN substrate has been added into the revised manuscript based on the review’s suggests. The corresponding revised parts has been marked red.

Reviewer 2 Report
In their revised version the authors have addressed all the points raised in my report. The paper can be accepted in the present form.
Author Response

(The authors gave the same response as above.)

Round 3
Reviewer 1 Report
For now, it appears that all of the issues raised from the previous reviews have been resolved. So I think the manuscript in the present form is in the state being acceptable for publication as it is after possible minor English proofreading.